



# sedExnerFoam 2412: A 3D Exner-based sediment transport and morphodynamics model

Matthias Renaud[1,3], Olivier Bertrand[3], Cyrille Bonamy[1], and Julien Chauchat[2]

[1]Univ. Grenoble Alpes, LEGI, CNRS, Grenoble INP
[2]Univ. Grenoble Alpes, INRAE, CNRS, IRD, Grenoble INP, IGE, Grenoble, France
[3]Artelia, 4 Rue Germaine Veyret-Verner, 38130 Échirolles, France

**Correspondence:** Matthias Renaud (matthias.renaud@grenoble-univ-alpes.fr)

**Abstract.** The development of an open source numerical model for sediment transport and morphological evolution is presented. It relies on the Arbitrary Lagrangian Eulerian (ALE) method to track the bed interface position over time. The sediment bed acts as a moving boundary whose motion depends on sediment fluxes and a dynamic mesh is employed to adapt the computational domain to the dynamic boundary. The implementation of the different components of the model (bedload, suspended transport, avalanche, etc.) is validated using a series of academic benchmarks. Finally, in order to highlight the model capability, an application to the study of a lone dune migrating under the influence of a steady flow is presented.

## 1 Introduction

The transport of sediments and morphodynamics that is, the evolution of the sedimentary bed is a complex physical problem involving many processes related to fluid mechanics, through the action of water on sedimentary particles, and solid mechanics when avalanches occur due to gravity. A coupling instability mechanism between fluid flow and bed evolution can also lead to the formation of bedforms, typically ripples or dunes (Kennedy, 1963; Charru et al., 2013). These bedforms alter the bed roughness and create a feedback loop on the fluid flow, which can result in a significant increase in flood risk in rivers or estuaries, for example (van der Sande et al., 2025; Hu et al., 2024). Therefore, morphodynamics models are essential tools for hydraulic engineers working on coastal, river, and estuarine systems, as they can be used to analyse erosion phenomena and assess the impact of human constructions, such as bridges, dams and renewable marine energy production systems (e.g. wind and tidal turbines).

The twentieth century saw the development of analytical models (Hjelmfelt and Lenau, 1970) and one-dimensional numerical models (Verwey, 1980; Goutal and Maurel, 2002). In the 1980s and 1990s, two-dimensional, depth-integrated and quasi-tridimensional numerical models emerged, primarily in the fluvial domain (Hervouet, 1999). Since the early 2000s, several three-dimensional (3D) models have been developed, including for coastal areas. Some are open-source, such as open-TELEMAC (Benoit et al., 2002), ROMS/CROCO (Warner et al., 2008; Marchesiello et al., 2015) and DELFT3D (Lesser et al., 2004), while others are proprietary, such as MIKE (Warren and Bach, 1992). Most of these models are adapted to flows on 'large spatial and temporal scales', and are often based on the use of sigma coordinates in the vertical direction. This does not allow for the integration of obstacles such as bridge piers or wind turbine masts (Hervouet, 2007; Lesser et al., 2004).



Another important approximation made in these models lies in the parametrization of the boundary layer: the first mesh point at the bottom is located in the logarithmic layer. Therefore these models are not particularly suitable for simulating interactions between morphodynamics and fluid flow around structures laid on the bottom, or for simulating processes such as scouring or bed instability, including the formation of ripples and dunes.

A new generation of 3D models based on emergent computational fluid dynamics (CFD) (Liu and García, 2008; Jacobsen,
2011; Baykal et al., 2015), allows for a finer resolution of flow and turbulence, particularly in the boundary layer and in the wake zones around structures. These models are based on a Lagrangian-Eulerian approach to handle the evolution of the bed boundary and the deformation of the associated volume mesh (ALE). To our knowledge, there is no open-source model of this type. While other approaches are possible, such as the immersed boundary method (IBM) (Song et al., 2022) or multiphase approaches (Chauchat et al., 2017; Nagel et al., 2020; Gilletta et al., 2024), these are too computationally expensive
for engineering applications. The ALE method therefore seems to be the best compromise. As part of a collaboration between the University of Grenoble Alpes (CNRS, Grenoble INP and INRAE) and the engineering company ARTELIA Group, an open source model is being developed within the C++ library OpenFOAM® (v2412) (Jasak et al., 2007). Named *sedExnerFoam*, this model is based on the ALE approach and was developed to meet the needs of hydraulic engineering. In particular, to provide a relevant tool for studying scour around hydraulic structures. However, the model's scope extends beyond this to include a wide
range of morphodynamics problems.

Scour is a specific sediment transport problem that requires fine local resolution in order to accurately capture the flow features around the obstacle (Song et al., 2022). To address this, the model relies on a CFD approach to solve the hydrodynamics and the excess of shear stress exerted on the sediment bed. This enables possible the study of various problems that cannot be simulated with depth integrated models or models that rely on boundary layer parametrization. For instance, the migration of
steep bedforms with flow separation occuring at their lee side due to the adverse pressure gradient (van der Sande et al., 2025), or scour around a bridge pile (see figure 1) and the horseshoe vortex which is the driving mechanism causing erosion upstream of the pile (Chiew and Melville, 1987; Roulund et al., 2005), or jet driven scour downstream of a sluice gate (Chatterjee et al., 1994; Martino et al., 2019).

Following an in-depth presentation of *sedExnerFoam* and its algorithm, the model undergoes rigorous validation using a
series of academic test cases. Finally, the model's potential applications are demonstrated by using it to study the migration of an isolated dune over a rigid bed in a steady flow.

## 2 Mathematical description

### 2.1 Hydrodynamics

The hydrodynamics is described by the incompressible filtered Navier-Stokes equations.

$$\frac{\partial \boldsymbol{u}}{\partial t} + \nabla.(\boldsymbol{u}^T \boldsymbol{u}) = -\frac{1}{\rho_f}\nabla p + \boldsymbol{g} + \nabla.(2\nu \boldsymbol{S} + \boldsymbol{\tau_f}),$$


$$\nabla.\boldsymbol{u} = 0,$$

(1)



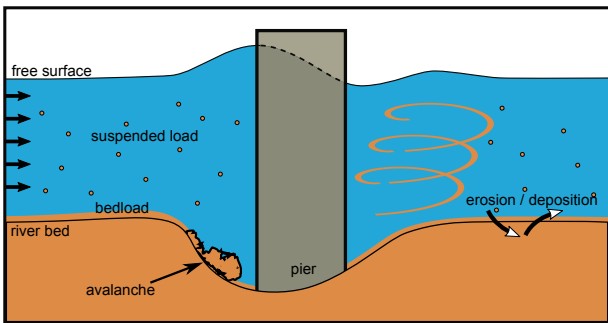

**Figure 1.** Physical processes involved in scour phenomenon.

where $\boldsymbol{u}$ is the fluid velocity, $p$ the fluid pressure, $\boldsymbol{g}$ the gravitational acceleration, $\rho_f$ the fluid density, $\nu$ the fluid kinematic viscosity and $\boldsymbol{S} = \frac{1}{2}[\nabla\boldsymbol{u} + (\nabla\boldsymbol{u})^T]$ is the strain rate tensor. $\boldsymbol{\tau_f}$ is a tensor which definition depends on the type of filter used. It can either be the specific Reynolds stress tensor to run Reynolds Averaged Simulations (RAS), a subgrid scale stress tensor when performing Large Eddies Simulations (LES) or the null tensor in case of laminar simulation or Direct Numerical
Simulation (DNS). The model makes use of the vast panel of possibilities offered by OpenFOAM and let the user choose freely the kind of filtering to be applied to equation 1. This is done by changing entries in a file *turbulenceProperties*. In this work however, the numerical simulations presented are either laminar cases (no filtering of the Navier-Stokes equations) or unsteady RAS simulations.

At this stage of model development, no feedback of the suspended load on the hydrodynamics is considered, which is a valid
assumption only in the case of dilute suspended sediments.

## 2.2    Turbulence modeling

As stated previously, in the case of RAS filtering the tensor $\tau_f$ from equation 1 is equal to the specific Reynolds stress tensor $\boldsymbol{\tau_f} = -\overline{\mathbf{u'^T}\mathbf{u'}}$. The use of the Reynolds stress tensor introduce 6 additional unknowns (the velocity fluctuation correlations) in the system of equations. The system as such is undetermined and the classical Boussinesq assumption is used as a closure. It
expresses the Reynolds stress tensor as a function of the eddy viscosity $\nu_t$ and $k = \frac{1}{2}\overline{\boldsymbol{u'}.\boldsymbol{u'}}$, the turbulent kinetic energy:

$$\boldsymbol{\tau_f} = 2\nu_t\boldsymbol{S} - \frac{2}{3}k\,\boldsymbol{I_3}\,, \tag{2}$$

where $\boldsymbol{I_3}$ is the identity matrix. Then a turbulence model is used to compute $\nu_t$. OpenFOAM offers multiple turbulence models to users, many of which are two-equations based on $k$, the turbulent kinetic energy (TKE) and either $\epsilon$ or $\omega$, the rate of dissipation of TKE and the specific rate of dissipation of TKE respectively. A transport equation is then solved for each
variable.

Of the various turbulence models available for the RAS approach in OpenFOAM ($k - \epsilon$, $k - \omega$, $RNG\,k - \epsilon$ ...), only the well-known $k - \omega$ Shear Stress Transport (SST) model is used in this work. It was first introduced by Menter (1994) and was





initially derived for aerodynamics study. The $k - \omega\, SST$ consists of a combination of two other classical turbulence models, the $k - \epsilon$ (Launder and Spalding, 1983) and the $k - \omega$ (Wilcox et al., 1998) models. The aim is to take the best out of those two

models. Indeed, the $k - \epsilon$ is known to work well for free shear flows but it performs poorly in the presence of adverse pressure gradients and is therefore not suitable for flows with boundary layer detachment. Conversely, the $k - \omega$ model is more effective in describing these types of flows but is less efficient than the $k - \epsilon$ in describing free shear flows in regions outside the range of influence of the solid boundaries (e.g. rigid walls, sediment bed). The $k - \omega\, SST$ model transitions between the two models using blending functions that take the distance to the nearest wall as input. The eddy viscosity $\nu_t$ is expressed as follows:

$$\nu_t = a_1 \frac{k}{\max(a_1\omega, b_1 F_2 S)}\,, \tag{3}$$

where $F_2$ is a blending function. The temporal evolutions of $k$ and $\omega$ are described by two transport equations:

$$\frac{\partial k}{\partial t} + \boldsymbol{u}.\nabla k = P - \beta^* k\omega + \nabla.((\nu + \sigma_k \nu_t)\nabla k)\,, \tag{4}$$

$$\frac{\partial \omega}{\partial t} + \boldsymbol{u}.\nabla\omega = \alpha S - \beta\omega^2 + \nabla((\nu + \sigma_\omega \nu_t)\nabla\omega)$$
$$+ 2(1 - F_1)\sigma_{\omega 2}\frac{1}{\omega}\nabla k.\nabla\omega\,, \tag{5}$$

where $F_1$ is another blending function. The different constants are obtained from the ones of the $k - \epsilon$ and the $k - \omega$ model using a blending function, $\alpha = \alpha_\omega F_1 + \alpha_\epsilon(1 - F_1)$, where $\alpha_\omega$ and $\alpha_\epsilon$ are constants from the $k - \omega$ and $k - \epsilon$ model respectively and $\alpha$ the corresponding constant of the $k - \omega\, SST$ model. The different blending functions $F_1$, $F_2$ and constants of the model are detailled in Menter et al. (2003).

### 2.3    Suspended sediment transport

In *sedExnerFoam*, the suspended load is described by the suspended sediment volume fraction $c_s = V_s/(V_s + V_f)$ where $V_s$ and $V_f$ stand for the volume of sediment and the volume of fluid, respectively. The evolution of $c_s$ in space and time is governed by an advection-diffusion equation:

$$\frac{\partial c_s}{\partial t} + \nabla.[(\boldsymbol{u} + \boldsymbol{w_s})c_s] = \nabla.(\epsilon_s \nabla c_s)\,, \tag{6}$$

where $\boldsymbol{w_s}$ is the sediment settling velocity and $\epsilon_s$ is the turbulent diffusivity for the suspended sediments. It is expressed as the

ratio of the turbulent eddy viscosity and the Schmidt number $\sigma_c$. The possibility to use an additional diffusivity $\epsilon_w$ in near bed areas is discussed later on. The suspended sediments concentration is supposed to behave has a passive scalar being transported with the flow and settling due to the effect of gravity. The settling velocity is computed as follows:

$$\boldsymbol{w_s} = w_s^0.F_h(c_s)\,\boldsymbol{e_g}\,, \tag{7}$$





| keyword | formula (for $C_d$ or $w_s^0$) | references |
|---|---|---|
| | terminal fall models | |
| Stokes | $w_s^0 = \frac{1}{18\nu}(s-1)gd^2$ | Stokes (1901) |
| Fredsoe | $C_d = 1.4 + \frac{36}{R_{ep}}$ | Fredsoe and Deigaard (1992) |
| Soulsby | $w_s^0 = \frac{\nu}{d}\sqrt{10.36^2 + 1.049D_*^3}$ | Soulsby and Whitehouse (1997) |
| Rubey | $w_s^0 = \left(\sqrt{2/3 + 36D_*^{-3}} - \sqrt{36D_*^{-3}}\right)\sqrt{(s-1)gd}$ | Rubey (1933) |
| fixedValue | value given by user | |
| | hindrance models | |
| Zaki | $F_h(C_s) = (1 - c_s)^n$ | Richardson and Zaki (1954) |
| ZakiModified | $F_h(c_s) = (1 - c_s)^{n-1}(1 - c_s/c_{s,max})$ | Camenen (2008) |
| fixedValue | value given by user | |

**Table 1.** Different available options in *sedExnerFoam* to compute the terminal falling velocity $w_s^0$ and hindrance functions $F_h$. Models are selected in the file *suspensionProperties* using the entries *fallModel* and *hindranceModel*.

where $w_s^0$ is the terminal sediment settling velocity of a single particle in a quiescent fluid, $\boldsymbol{e_g} = \boldsymbol{g}/|\boldsymbol{g}|$ is a unit vector oriented with gravity, and $F_h$ is an hindrance function that takes values between 0 and 1 and is a decreasing function of $c_s$. It represents the effect of particles hindering each other as they fall leading to a drop of their settling velocity as $c_s$ increases (Richardson and Zaki, 1954). The different models available to compute the terminal falling velocity $w_s^0$ and the hindrance function $F_h$ are summarized in table 1.

The values taken by the Schmidt number $\sigma_c$ has been a hot topic up to today without a consensus being reached. van Rijn (1984) proposed a formula to estimate $\sigma_c$ from the settling velocity and the friction velocity $u_*$:

$$\sigma_c = \frac{1}{1 + 2\left(\frac{w_s}{u_*}\right)^2}, \quad \text{for} \quad 0.1 < \frac{w_s}{u_*} < 1. \tag{8}$$

This yields a Schmidt number smaller than one which corresponds to sediment diffusion being more intense than the turbulent diffusion for the fluid. Others tried to experimentally estimate $\sigma_c$ and highlighted the non uniformity of the Schmidt number (Amoudry et al., 2005). However, this issue has yet to be solved and the Schmidt number is assumed constant in the present model and its value is set by the user.

The final key aspect of this approach is how to enforce the bed boundary condition, that is the exchange of mass between the sediment bed and the suspended load. This topic is covered at the end of the next section in relation to bedload and morphological evolution.





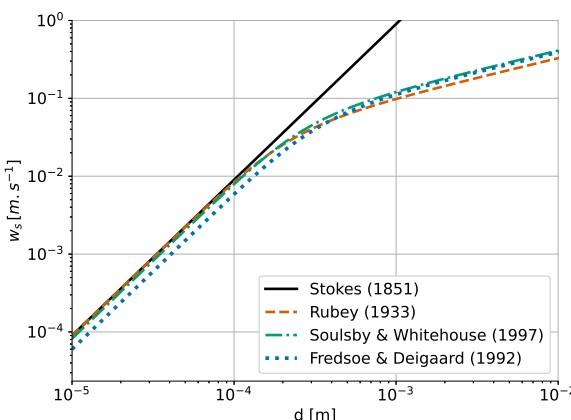

**Figure 2.** Different formulas to compute the terminal settling velocity of a sand particle ($\rho_s \approx 2650\,kg.m^{-3}$) as a function of its diameter.

## 2.4 Bedload and Morphodynamics

### 2.4.1 Exner equation

The morphological evolution of a granular bed in a wide range of sediment transport problems is modeled by the so-called Exner equation. It was first proposed by Exner (1920), an Austrian meteorologist and geophysicist. In their article, Paola and Voller mention that Felix Exner initially suggested that the bed elevation was evolving proportionally with the divergence of the mean flow velocity but made clear that the mean flow acted as a proxy for the sediment flux. This led to the standard
formulation, which written as follows:

$$(1 - \lambda_s)\frac{\partial z_b}{\partial t} + \nabla_H.\boldsymbol{q_b} = D - E\,, \tag{9}$$

where $z_b$ is the bed elevation, $\lambda_s$ is the porosity of the granular material which is linked to the maximum possible sediment volume fraction $c_s^{max} = 1 - \lambda_s$. The bedload flux $\boldsymbol{q_b}$ is the specific flux of sediment transported along the bed per unit width. It is computed from the bed shear stress using an empirical formula. The formulas available to the user are all summarized in
table 2 and discussed in the next section. $D$ and $E$ are respectively the deposition and erosion flux, they are the terms through which sediment is exchanged between the bed and the water column. The Exner equation is a 2 dimensional equation and is thus solved after applying a 2 dimensional plane projection on all variables. The operator $\nabla_H$ stands for the divergence operator on this projected plane.

### 2.4.2 Bedload modeling

In the 1930's, Albert Frank Shields made measurements of the motion threshold already highlighted by Du Boys in 1879. The particles start to move when the Shields number $\theta = \frac{\tau_b}{(\rho_s - \rho_f)gd}$ exceeds a critical value $\theta_c$, where $\tau_b$ is the shear stress exerted



| keyword | formula | references |
|---------|---------|------------|
| | critical Shields number | |
| Brownlie | $\theta_c^0 = \frac{0.22}{D_*^{0.9}} + 0.06\,10^{-7.7D_*^{-0.9}}$ | Brownlie (1983) |
| Miedema | $\theta_c^0 = \frac{0.2285}{D_*^{1.02}} + 0.0575(1 - e^{-0.0225D_*})$ | Miedema (2008) |
| Soulsby | $\theta_c^0 = \frac{0.3}{1+1.2D_*} + 0.055\left(1 - e^{-0.02D_*}\right)$ | Soulsby and Whitehouse (1997) |
| Zanke | $\theta_c^0 = \frac{0.145}{D_*^{0.5}} + 0.045\,10^{-1100D_*^{-2.25}}$ | Zanke (2003) |
| | bedload transport formulas | |
| Camenen | $\phi_b = 12\,\theta^{1.5}e^{-4.5\theta_c/\theta}$ | Camenen and Larson (2005) |
| MeyerPeter | $\phi_b = 8\,\varpi(\theta - \theta_c)^{3/2}$ | Meyer-Peter and Müller (1948) |
| Nielsen | $\phi_b = 12\,\theta^{1/2}\varpi(\theta - \theta_c)$ | Nielsen (1992) |
| vanRijn | $\phi_b = 0.053\frac{\varpi(\theta/\theta_c-1)^{2.1}}{D_*^{0.3}}$ | Van Rijn (1984) |
| custom | $\phi_b = \eta_b\theta^a\varpi(\theta - \theta_c)^b$ | |

**Table 2.** Available formulas to compute the critical Shields number from the fluid and sediments physical properties and formulas to compute the bedload flux from the Shields number. The bedload flux is rendered dimensionless by the Einstein number. Those formulas are selected in the file *bedloadProperties* using the entries *criticalShieldsModel* and *bedloadModel*.

by the flow on the bed. $\rho_s$ and $\rho_f$ are the density of the sediments and the fluid respectively. In his work, Albert Shields showed that the critical Shields number is Reynolds dependent leading to the development of various empirical formulas trying to estimate $\theta_c$. Different formulations based on the dimensionless sediment particle diameter $D_*$ have also been proposed in the literature such as in the work of Soulsby and Whitehouse (1997) and Brownlie (1983).

$$D_* = d\left(\frac{(s-1)g}{\nu^2}\right)^{1/3} \tag{10}$$

The various formulas available in the model are summarized in table 2 and represented on figure 3. The user can choose between one of those models or manually set a value for $\theta_c$.

Accurately measuring the threshold of motion is a difficult task mainly because of the absence of a universal definition of the motion threshold. Some particles can indeed be seen moving even for subcritical values of $\theta$. Furthermore, corrections need to be applied to the value of $\theta_c$ to account for bed slope effect. In the following, the base critical Shields number, which is the critical Shields number on a flat bed, is noted $\theta_c^0$. The critical Shields number after slope correction is denoted by $\theta_c$. Following



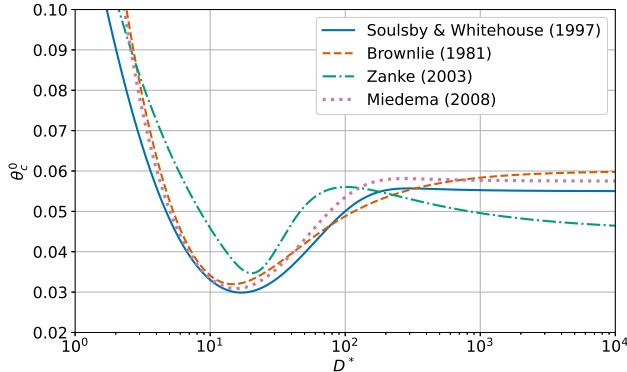

**Figure 3.** Critical Shields number as a function of the dimensionless sediment diameter $D_*$.

Fredsoe and Deigaard (1992), a correction to account for the local bed slope is applied to the critical Shields number:

$$\frac{\theta_c}{\theta_c^0} = \cos(\beta_s)\sqrt{1 - \frac{\sin(\alpha_s)^2 \tan(\beta_s)^2}{\mu_s^2}} - \frac{\cos(\alpha_s)\sin(\beta_s)}{\mu_s}, \tag{11}$$

where $\beta_s$ is the angle of the bed slope and $\alpha_s$ the angle between the steepest slope direction and the direction of the shear. The coefficient of static friction $\mu_s$ is linked to the angle of repose of the granular material $\beta_r$ through $\tan(\beta_r) = \mu_s$. The slope correction can be activated/deactivated in the file *bedloadProperties* using the entry *slopeCorrection*.

Various studies have focused on trying to find relationships between the Shields number and $\phi_b = |\boldsymbol{q_b}|/\sqrt{(s-1)gd^3}$ (Einstein, 1942; Meyer-Peter and Müller, 1948; Van Rijn, 1984), the dimensionless bedload flux leading to the development of numerous empirical relations. A lot of those formulas are of the form:

$$\phi_b \propto \theta^a \varpi (\theta - \theta_c)^b, \tag{12}$$

with $a$ and $b$, two real positive coefficients. The formulas available for the user to compute the bedload are summarized in table 2 and plotted in figure 4. The user also has the possibility to define a custom bedload formula by manually setting the prefactor and the coefficients $a$ and $b$ in equation 12.

Another phenomenon that need to be considered when solving the Exner equation, are the sediments avalanches occurring when the bed slope exceeds the angle of repose of the granular material. If not taken into account, unrealistic slopes could appear in the numerical solution or even shock situations which could trigger numerical instabilities in the model. Marieu et al. (2008) proposed a model based on an iterative procedure to redistribute the excess of sediment locally until the bed slope does not exceeds the granular material angle of repose. Such a procedure has been successfully tested in other works such as Zhou (2017). In *sedExnerFoam* however, the avalanche is modeled with an additional bedload term $\boldsymbol{q_{av}}$ inspired from Duran Vinent et al. (2019):

$$|q_{av}| = q_{av}^0 \frac{\varpi[\tanh(\tan(\beta_s)) - \tanh(\tan(\beta_r))]}{1 - \tanh(\tan(\beta_r))}, \tag{13}$$



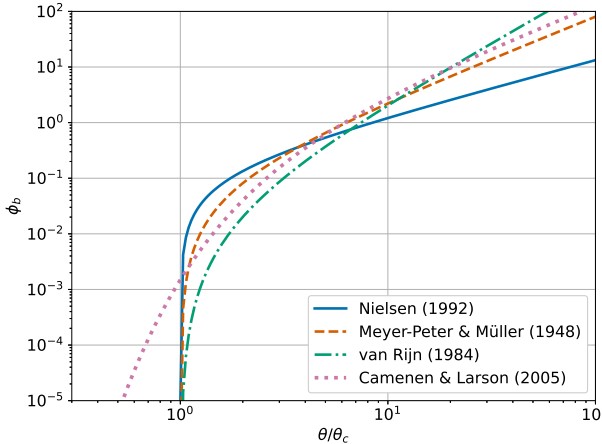

**Figure 4.** The adimensioned bedload flux as a function of the relative Shields number $\theta/\theta_c$ from the empirical formulas detailed in table 2.

with $\beta_s$ and $\beta_r$ respectively the angle of the slope and the repose angle of the granular material. $q_{av}^0$ is a positive constant which can be set by the user. It corresponds to the maximum possible additional bedload flux due to the avalanche. This avalanche flux is oriented toward the steepest slope direction. One benefit of this formulation is that it enables slopes to exceed the angle of repose in the event of competition between bedload flux due to bed shear stress and that due to gravitational acceleration.

One last aspect which is often neglected when modeling sediment transport in water but is widely used in eolian transport is the saturation of the bedload flux. From Charru (2006); Charru et al. (2013) the saturation can be expressed in a coordinates system aligned with the shear stress as:

$$T_{sat}\frac{\partial q_b}{\partial t} + L_{sat}\frac{\partial q_b}{\partial x} = q_{sat} - q_b\,, \tag{14}$$

where $q_{sat}$ is the saturated flux, $T_{sat}$ the saturation time, $L_{sat}$ the saturation length and $x$ is the coordinate in the direction of the shear stress. From the file *bedloadProperties*, the user can activate the saturation and provide values for $T_{sat}$ and $L_{sat}$. The saturated flux $\boldsymbol{q_{sat}}$ is then computed from the bed shear stress using one of the formula from table 2 and the bedload flux $\boldsymbol{q_b}$ is the solution of equation 14.

### 2.4.3 Erosion and deposition fluxes

As said previously, the modeling of the erosion and deposition fluxes is a hard point of classical sediment transport models. Many sediment transport experiments in straight flumes have been conducted to study the relation between the flow and the rate at which particles are eroded from the bed to the water column. van Rijn (1984) studied the case of sediment transport in a straight channel under equilibrium condition and proposed an empirical formula to compute a reference concentration $c_b^*$ at a certain reference distance from the bed, the so-called reference level $\delta z_b^*$:



$$c_b^* = 0.015 \frac{d}{\delta z_b^*} \frac{(\theta/\theta_c - 1)^{3/2}}{(D_*)^{0.3}} . \tag{15}$$

The reference concentration corresponds to the concentration observed at a distance $\delta z_b^*$ from the bed under equilibrium condition.

Since then, a lot of sediment transport models used van Rijn empirical formula and assumed an equilibrium situation at the reference level to impose the boundary condition $c_s(\delta z_b^*) = c_b^*$. However, this boundary condition is not appropriate for a lot of situation where a local equilibrium cannot be assumed at the reference level. Celik and Rodi (1988) adapted this boundary condition to handle out of equilibrium situation. The erosion flux is written $E = w_s c_b^*$ and the deposition flux $D = w_s c_b$ with $c_b$ a sediment concentration value computed from the values in the neighbouring cells which is detailed later on. The idea is that the rate of erosion is always the same as the one under equilibrium condition and the deposition depends only on the concentration value in the few first cells above the bed. If $c_b > c_b^*$, then suspended sediment get deposited on the bed and when $c_b < c_b^*$, sediment get eroded from the bed and suspended in the water column. The equilibrium occurs when $c_b = c_b^*$.

One difficulty is then how to impose this reference concentration at the reference level which is above the bed boundary. Large scale sediment transport models avoid this difficulty by not meshing the region located in between the sediment bed and the reference level. The downfall of this method being that the flow near the bed is not solved and need to be modeled, typically leading to a bad hydrodynamics in highly non uniform flow regions such as near obstacles. In order to maintain a good hydrodynamics resolution Jacobsen (2011) developed a model relying on a different mesh for the hydrodynamics and for the suspended load. The mesh for the suspended load bottom boundary was located at the reference level whereas the mesh for the hydrodynamics presented cells in between the sediment bed and the reference level. In *sedExnerFoam* it was chosen to avoid the use of two different meshes.

As stated previously, the deposition and erosion fluxes are computed as suggested by Celik and Rodi (1988). The erosion $E$ is computed at the reference level $\delta z_b^* = k_s$, the Nikuradse equivalent roughness height ($k_s = 2.5d$), using equation 15 and a limiter so that $c_b^*$ is not exceeding a value $c_{b,max}^*$, typically equal to half the maximum possible sediment volume fraction. This limiter is needed to avoid $c_b^*$ taking non physical values when the bed shear stress becomes important (see eq. 15). In their work Amoudry et al. (2005) use a maximum possible reference concentration $c_{b,max}^* = 0.3$ which is close to the value of 0.32 suggested by Engelund and Fredsøe (1976). The computed reference concentration is then extrapolated at the height of the first cell center above the sediment bed $z_1$ using the formula suggested by Fang and Rodi (2003):

$$c_{b1}^* = min\left( c_b^* e^{-\frac{w_{s1}}{\epsilon_{s1}}(z_1 - \delta z_b^*)}, c_{b,max}^* \right) , \tag{16}$$

where $c_{b1}^*$ is the reference concentration extrapolated at the height $z_1$. $w_{s1}$ and $\epsilon_{s1}$ stand for the settling velocity and the sediment turbulent diffusivity values at the center of the first cell above the sediment bed located at a height $z_1$. The expression of $c_{b1}^*$ is obtained by considering a local equilibrium in a small region above the bed and assuming $\epsilon_s$ and $w_s$ to be uniform between the reference level $\delta z_b^*$ and the center of the first cell above the bed. The deposition and erosion are then computed at





the first cell center and not on the bed boundary, leading to $D = w_s c_1$. The total erosion/deposition flux is then estimated as:

$$D - E = w_{s1}(c_1 - c_{b1}^*).\tag{17}$$

This flux is imposed as a boundary condition for the suspended-load transport (equation 6). With this method the same mesh
can be used for both the suspended load and the hydrodynamics with a fine resolution near the sediment bed. It is to be remembered that the various formulas for $c_b^*$ existing in the literature are all empirical and are based on measurements made in straight channel flow experiments. Their validity out of this configuration, let alone in the vicinity of an obstacle disturbing the flow, is subject to caution.

The possibility to use an additional diffusivity near the bed for suspended sediments as been introduced in the model after
225 experimenting difficulties to suspend material from the bed to the water column in the case of fine grid resolution and low to medium dimensionless roughness $k_s^+ = \frac{k_s u_*}{\nu}$. The eddy viscosity being very low on the first cells above the bed, the eroded sediment stay in the first layer of cells above the bed without rising up in the water column. To solve this issue an additional artificial diffusivity is introduced in the near bed region:

$$\frac{\epsilon_w}{\nu} = \frac{\epsilon_w^0}{2}\left(1 - \tanh\left(\xi_w \frac{z - k_s}{k_s}\right)\right).\tag{18}$$

It can be interpreted as a saltation effect. In the presence of a viscous sublayer above the bed, the particles need to reach a certain elevation in order to get caught by turbulent eddies and rise in the water column. If the flow is rough however ($k_s^+ \gg 1$), the turbulence reaches the bed and the use of $\epsilon_w$ is not needed. The coefficients $\epsilon_w^0$ and $\xi_w$ are both set to 5 by default but can be modified in the file *bedloadProperties*.

## 3 Numerical implementation

### 3.1 Code implementation

The numerical implementation of *sedExnerFoam* is based on the finite volume method (FVM) using OpenFOAM® (v2412). The Navier-Stokes equations (eq. 1) and the transport equation for suspended-load (eq. 6) are both solved using the finite volume method. The computational domain is split into a multitude of discrete polyhedral control volumes over which the partial differential equations are integrated.

The Exner equation, however, is solved over a surface (the sediment bed) using the finite area method (FAM). FAM is an adaptation of the finite volume methods on a surface curved in the 3 dimensional space. It was initially developed by Tukovic and Jasak (2008) for the numerical study of the transport of a surfactant at the interface between two fluids and has since then been successfully applied to other problems such as dense-flow avalanches (Rauter and Kowalski, 2024). In the present model, the finite area mesh is mixed with the patch of the volumic mesh corresponding to the sediment bed. The partial differential
equations discretization with the finite area method was initially developed to take into account the curvature of the surface, however no curvature effect is taken into account for the bed morphology evolution. The Exner equation is solved on a projected plane normal to the gravity vector $\boldsymbol{g}$.





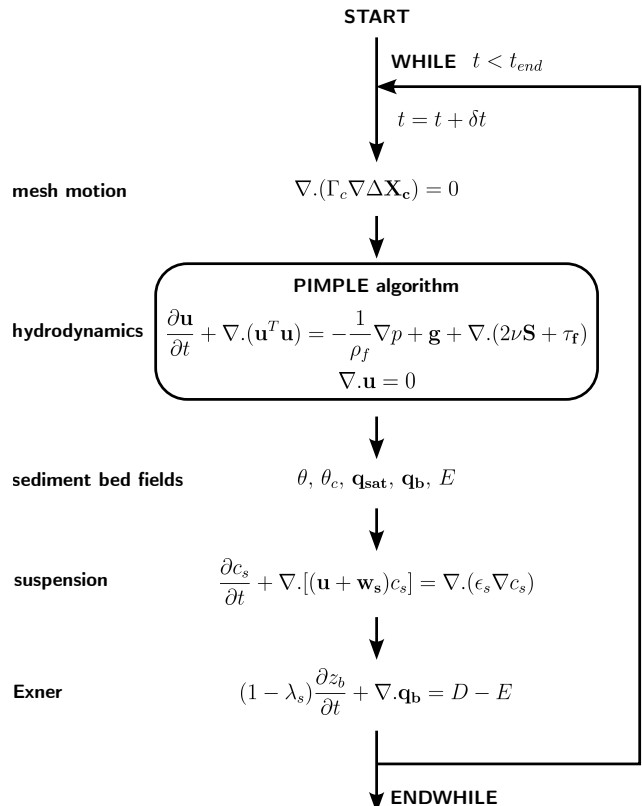

**Figure 5.** Flow chart of *sedExnerFoam*.

The sequence of operations performed during a time iteration and their sequence is represented on figure 5. After solving the mesh deformation, the differential equations for the velocity field $\mathbf{u}$, the pressure $p$ (eq. 1) as well as transport equations

for fields related to turbulence modelling (eq. 4, 5) are first solved through the PIMPLE algorithm for transient solution which is detailed in Greenshields and Weller (2022). It consists in a mix of the SIMPLE (Semi-Implicit Method for Pressure-Linked Equations) from Patankar and Spalding (1983) and the PISO (Pressure Implicit with Splitting of Operators) from Issa (1986). An additional corrector loop called PIMPLE loop is added above the PISO loop. During a time step, the velocity flux through the mesh faces to be updated at each PIMPLE loop iteration, preserving the simulation stability at higher Courant number

($C_o > 1$). The PISO algorithm behavior is restored by disabling the PIMPLE loop. Once the hydrodynamics has been solved, the shear stress exerted on the bed is computed as well as the associated bedload and erosion flux. The transport equation for suspended sediment transport is solved and the deposition flux is deduced from it. Lastly the bed boundary motion is computed by explicitly solving the Exner equation. At the begining of the next time iteration, the mesh is updated to match the new bed position.





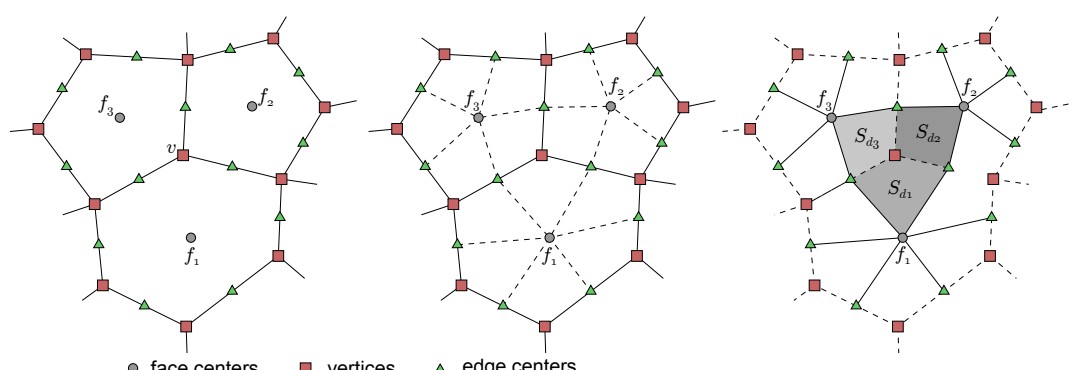

● face centers   ■ vertices   ▲ edge centers

**Figure 6.** Decomposition of the horizontal projection of the finite area mesh into a dual mesh.

## 3.2 Exner equation resolution

Let us integrate the Exner equation (eq. 9) over the projection of a face $f$:

$$\left.\frac{\partial z_b}{\partial t}\right|_f = -\frac{1}{S_{fp}} \sum_e (\boldsymbol{q_{b,ep}}.\boldsymbol{n_{ep}})l_{ep} + (D-E)_f, \tag{19}$$

where $S_{fp} = S_f(\boldsymbol{n_f}.\boldsymbol{e_g})$ is the projected area of face $f$ and $S_f$ is face $f$ area, $\boldsymbol{n_f}$ is the face normal unit vector oriented outward of the computational domain and $\boldsymbol{e_g}$ is a unit vector oriented along the gravity vector. $l_{ep}$ is the length of the projected edge and $\boldsymbol{n_{ep}}$ the projected normal edge vector which is oriented toward the outside of $f$. The users have the choice to use either a explicit Euler scheme or a first order Adams-Bashforth scheme, which is a second order time scheme, for temporal discretization.

From equation 19, at each time step, an increment of bed elevation $\delta z_b$ is computed for each face centers. In OpenFOAM, the mesh geometry is defined by the vertices coordinates. Thus, to impose a mesh motion, the vertical displacements computed at face centers need to be interpolated on the vertices. Particular caution must be given to the interpolation scheme to ensure mass conservation. A naive approach would be to linearly interpolate $\delta z_b$ from face centers to vertices, which would be mass conservative for cases limited to one horizontal dimension and structured mesh but would fail to preserve the mass in the general case of an unstructured mesh. Jacobsen (2015) made a detailed review of the different possible methods to solve the Exner equation and analysed their benefits and shortcomings. He proposed a mass conservative interpolation scheme which is the one implemented in the current model.

A dual mesh is constructed as depicted in figure 6. Each vertex $v$ belonging to the primary mesh is the center of a face of the dual mesh. The vertices delimiting this dual face are the neighbouring primary faces $f_i$ sharing the vertex $v$ as well as the centers of the edges whose one end is vertex $v$. The mass increment of the sediments contained under a face $f$ is $m_f = \rho_s S_{fp} \delta z_b|_f$, where $\delta z_b|_f$ is the face elevation increment. To ensure mass conservation during the interpolation process, the sum of the mass contained under every faces must be the same when computing this sum for the initial mesh and for the





dual mesh. The vertical displacement of each vertex $\delta z_b|_v$ is then a linear combination of the displacements of the faces sharing this vertex. The weight associated with each face is proportional to the area of the quadrilateral defined by the face center, the vertex and the centers of the two edges belonging to the face and sharing the vertex (see figure 6). Let us note the area of this quadrilateral $S_{df}$, the value of the elevation increment $\delta z_b|_v$ associated to a vertex $v$ is computed:

$$\delta z_b|_v = \frac{1}{S_v} \sum_f S_{df} \delta z_b|_f \,, \tag{20}$$

where $S_v$ is the area of the dual face whose center is the vertex $v$. It is equal to the sum of the area of each face associated quadrilateral: $S_v = \sum_f S_{df}$. Thus, the sum of the interpolation weights is equal to 1. This interpolation method is mass conservative and also has the advantage of acting as a filter on the vertical bed displacement and helps to keep the numerical solution of the Exner equation stable.

### 3.3 Mesh motion

At each time step, solving the Exner equation gives a displacement for the bed boundary of the finite volume mesh. In order for the finite volume mesh to adapt to the bed boundary motion and to preserve the mesh quality throughout the simulation, a mesh motion solver based on a laplacian equation for cell center displacements is used:

$$\nabla.(\Gamma_c \nabla \boldsymbol{\Delta X_c}) = 0 \,, \tag{21}$$

where $\Gamma_c$ is the mesh diffusivity and $\boldsymbol{\Delta X_c}$ is the displacement of the cell centers. Solving equation 21, new positions of the mesh cell centers are obtained. The mesh vertices new coordinates are then interpolated from $\boldsymbol{\Delta X_c}$. The model is currently compatible with two mesh motion solvers which are selected in the file *constant/dynamicMeshDict*:

- displacementLaplacian

- displacementComponentLaplacian

Using a spatially non uniform mesh diffusivity $\Gamma_c$ allows to select regions where maintaining mesh quality and cell sizes is a priority. In regions where $\Gamma_c$ is lower, cells would get easily distorted. On the contrary, regions with higher values of $\Gamma_c$ would avoid cells shrinking or expanding if the bed boundary motion is not too large. Once again the user has multiple possibilities to set the values taken by $\Gamma_c$. The best practice is to maintain a high mesh diffusivity close to the sediment bed boundary to preserve the mesh quality in the near bed region. The user can select one of the following options in the file
*constant/dynamicMeshDict*:

- inverseDistance: $\Gamma_c = 1/L_{sb}$

- quadratic inverseDistance: $\Gamma_c = 1/L_{sb}^2$

- exponential: $\Gamma_c = e^{-L_{sb}}$





where $L_{sb}$ is the distance to the sediment bed boundary.

One drawback of the finite volume method to solve mesh motion, is the need for interpolation to get vertex displacements from cell centers displacements which is the results of equation 21. This interpolation step can lead to a drop in mesh quality in the regions where the bed motion is highly non uniform. In the worst case scenario, some cells in the domain could collapse, leading to the simulation failure.

Among all the simulations tried, one problematic case has been identified which is the migration of a steep bedform. If the
crest is sharp, then the vertices located just above the crest but not belonging to the bed boundary could be displaced below the bed boundary during the interpolation process. A reduction of the near bed cells aspect ratio is one effective way to get rid of this problem.

In their work, Jasak and Tukovic (2006) discussed in more details the issues arising when using a mesh motion procedure based on the finite volume method. They proposed a vertex-based method to solve the mesh motion, which avoid cell collapsing
problems. In order to apply this method on a mesh made up of arbitrary polyhedra, a decomposition of every polyhedron in a sum of tetrahedra is used. The downfall of this method is that equation 21 is then solved on a refined tetrahedral mesh composed of more cells than the initial mesh. As an example, applying this decomposition on an hexahedral mesh multiplies the number of cells by six. For that reason and because in the practice, the classical finite volume method gave satisfactorily results, the discussed vertex-based method is currently not available in *sedExnerFoam*. However, it could be implemented in the future if
it proves necessary for other studies.

## 4    Model validation

A series of tests is presented both to illustrate the model behavior of *sedExnerFoam* and to validate it against either analytical solutions or experimental results. The tests are chosen to isolate one physical process at a time. They are organised as follows: two tests involving suspended load transport only are first presented. Then the case of an idealized dune transport problem
for which an analytical solution exists is investigated. At last, the conservation of mass is illustrated by means of two tests on suspended sediment deposition and avalanches.

### 4.1    Suspension under equilibrium condition

A classical test is the suspension of sediment in a straight flume under equilibrium condition which has been extensively studied (van Rijn, 1984; Lyn, 1988; Muste et al., 2005). The situation is the following, a fully developed flow in a channel
is considered. The channel is supposed long enough so that the vertical profiles of velocity and turbulent eddy viscosity are stationary. Under equilibrium condition, the vertical profile of suspended sediment concentration is the results of a balance between the gravity which makes the particles to settle at a velocity $w_s$ and the mixing induced by turbulence. The transport equation for the suspended load (eq. 6) then reduces to:



| experiment | 1565 | 1965 | 2565 | 1957 |
|---|---|---|---|---|
| $d\,(cm)$ | 0.15 | 0.19 | 0.24 | 0.19 |
| $w_s\,(cm.s^{-1})$ | 1.6 | 2.3 | 3.1 | 2.3 |
| $\overline{u}\,(m.s^{-1})$ | 0.649 | 0.671 | 0.744 | 0.672 |
| $u_*\,(cm.s^{-1})$ | 3.58 | 3.75 | 4.25 | 3.95 |
| $R_o$ | 1.09 | 1.24 | 1.38 | 1.17 |

**Table 3.** Parameters of Lyn (1988) experiments. Particles diameter $d$, settling velocity $w_s$, mean water velocity $\overline{u}$, friction velocity $u_*$ and Rouse number $R_o$.

$$\frac{d}{dz}\left(-w_s c_s + \epsilon_s \frac{dc_s}{dz}\right) = 0\,. \tag{22}$$

Depending on the shear stress exerted on the bed, granular material is eroded and suspended in the water column. Then, the turbulent diffusion uplifts the particles until an equilibrium is reached. Assuming a parabolic turbulent viscosity profile, $\nu_t(z) = u_* \kappa z (H - z)$, the solution of equation 22 between the reference level $\delta z_b$ where the concentration is the reference concentration $c_b^*$, and the top of the water column $H$ is the so-called Rouse profile:

$$c_s(z) = c_b^*\left(\frac{H-z}{z}\frac{\delta z_b}{H - \delta z_b}\right)^{R_o}, \tag{23}$$

where $R_o = \sigma_c w_s / \kappa u_*$ is the Rouse number and $\kappa = 0.41$ is the von Kármán constant.

To validate the model, numerical results are compared with experimental data from Lyn (1988). The experiment was conducted in a 13 meters long and 26.7 centimeters wide flume with a bottom covered by a layer of sand. Flow and suspended sediments concentration measurements were made approximately 9 meters downstream of the channel entrance. The experiment parameters are summarized in table 3. For the four experiments, measurements of the velocity field, the velocity correlation and the suspended sediment concentration profiles are available.

The four equilibrium bed experiments are reproduced numerically. The mesh used is 1 dimensional. It consists in a column of 100 to 120 cells oriented in the z-direction and with cyclic boundary conditions in the x-direction. As a results, only the x-component $u$ of the velocity field is not null. For the 4 simulations, the $k - \omega\,SST$ turbulence model is used and the mesh resolution near the bed is kept to $z^+ \approx 1$ to ensure a good resolution and a correct estimation of the shear stress exerted on the bed.

The free surface is not considered, instead a rigid lid is applied at the top, with zero gradient condition for the turbulent kinetic energy $k$, a Dirichlet condition for $\omega$ and a slip boundary condition for the velocity $u$. To take into account the bed roughness effect on the hydrodynamics, the boundary condition for $\omega$ proposed by Wilcox et al. (1998) is used:

$$\omega = \frac{u_*^2}{\nu} S_R\,, \tag{24}$$



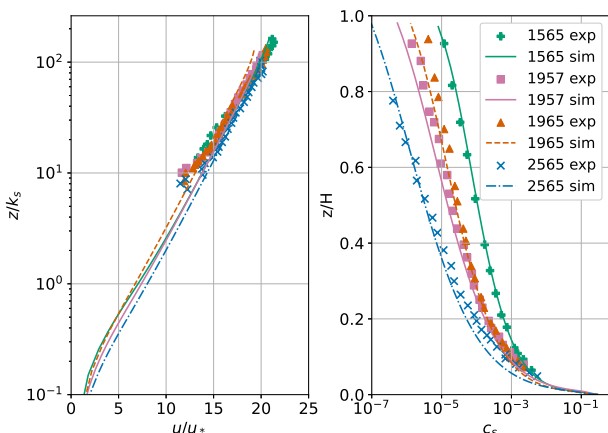

**Figure 7.** Velocity and suspended sediment concentration profiles from simulations and comparison with experimental data from Lyn (1988).

where $S_R$ is defined as a function of the roughness Reynolds number $k_s^+ = u_* k_s / \nu$ as follows:

$$S_R = \left( \frac{200}{k_s^+} \right)^2, \qquad \text{for } k_s^+ \leq 5 \tag{25}$$

$$S_R = \frac{100}{k_s^+} + \left[ \left( \frac{200}{k_s^+} \right) - \frac{100}{k_s^+} \right] e^{5-k_s^+}. \qquad \text{for } k_s^+ > 5 \tag{26}$$

The transient problem is solved and the simulations are run until a steady state has been reached. The numerical results are plotted alongside the experimental data from Lyn (1988) on figure 7.

For all 4 simulations, the turbulent Schmidt number was set to values slightly above 1, $\sigma_c \in [1.1, 1.2]$, meaning that the turbulent diffusivity for sediment concentration is lower than the turbulent diffusivity for the fluid. In general the numerical results show good agreement with the experiments. For the suspension profiles, it is observed that for the case 2565, not enough sediment is suspended in the lower part of the water column compared with the experiment. A better fit could be obtained by further adjusting the parameters $\epsilon_w^0$ and $\xi_w$ in equation 18 which are both set to 5. Other adjustment parameters are the turbulent Schmidt number and the bed roughness height $k_s$ considered in the boundary condition for $\omega$.

### 4.2 Suspension development

Another test for the suspended load is the development of suspension in a channel, when the flow encounters an abrupt transition from a non erodible bed to an erodible bed. Initially, the flow is clear, and it becomes loaded with sediments until an equilibrium is reached. For this problem, the results are compared with a pseudo-analytical solution derived by Hjelmfelt and Lenau (1970). In order to obtain this solution, some hypothesis are made.

1. The sediment is uniformly advected at the mean flow velocity $\overline{u}$.

2. The turbulent viscosity vertical profile is assumed parabolic, $\nu_t = \kappa u_* z (1 - z/H)$ for $z \in [\delta z_b, H]$ where $\delta z_b$ is the reference level and $H$, the water depth.





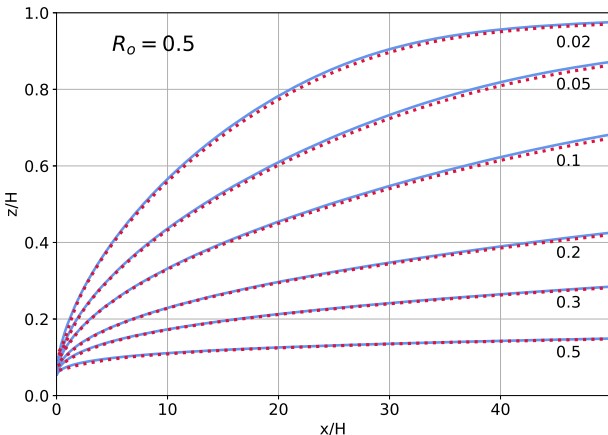

**Figure 8.** Isolines of $c_s/c_b^*$ for a Rouse number of 0.5 with hypothesis from Hjelmfelt and Lenau (1970) enforced except the null horizontal turbulent diffusion. Solid blue curves represent the model results and the dotted red ones the pseudo analytical solution.

3. The concentration at $z = \delta z_b$ is supposed to be constant along the flume and equal to $c_b^*$, the reference concentration.

4. The horizontal turbulent diffusion is neglected.

Based on those hypothesis, Hjelmfelt and Lenau (1970) simplified the transport equation for the suspended-load and derived an analytical solution. They performed a separation of variable and used the Sturm Liouville theory to obtain a solution which only depends on the Rouse number. A first numerical simulation is performed for which all assumptions apart from the fourth one are respected. A water depth of $H = 0.1\,m$ is considered, the mean velocity is $\overline{u} = 0.9\,m.s^{-1}$ and the Rouse number is

equal to 0.5 which corresponds to a highly suspended regime. The results are presented in figure 8.

In this case, the numerical and pseudo-analytical solutions are almost identical suggesting that the stream-wise turbulent diffusivity (hypothesis 4) is indeed negligible. However, some of the hypothesis from Hjelmfelt and Lenau (1970) are normally not verified. The vertical velocity profile is not uniform, the concentration at the reference level may vary in space and reach an equilibrium after some distance from the inlet and last the turbulent eddy-viscosity profile is not exactly parabolic (see

figure 10). A series of particle diameter values was selected to vary the Rouse number but here only the case $R_o = 0.5$, obtained for a particle diameter $d = 0.12\,mm$ and a settling velocity $w_s = 0.773\,cm.s^{-1}$, is presented. The same mean flow velocity $\overline{u} = 0.9\,m.s^{-1}$ is taken and the resulting shear stress exerted on the bed corresponds to the bed friction velocity $u_* = 3.77\,cm.s^{-1}$. The $k - \omega\,SST$ turbulence model and the rough wall boundary from Wilcox et al. (1998) (eq. 24) condition is used for $\omega$ with a roughness height $k_s = 2.5\,d$.

A first 1D simulation is performed without sediment to obtain vertical profiles for $u$, $k$ and $\omega$ corresponding to a fully developed channel flow. The fields $u$, $k$ and $\omega$ are extracted from this first simulation and used as the inlet boundary condition for the second simulation for which suspension is activated. The flow entering the domain being already fully developed, only $c_s$ vary with the x-position. The mesh consists in a 2-dimensional structured mesh more refined close to the bed to ensure





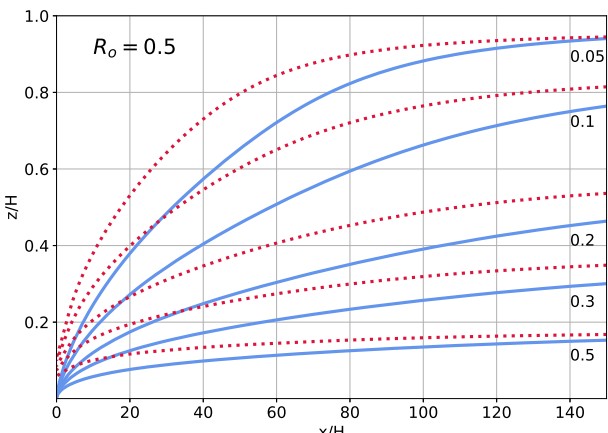

**Figure 9.** Isolines of $c_s/c_b^*$. Comparison between pseudo-analytical solution from Hjelmfelt and Lenau (1970) (red dotted lines) and the model solution (solid blue lines) without enforcement of the hypothesis used to derive the pseudo-analytical solution.

the condition $z^+ \approx 1$ ($n_x = 2000$, $n_z = 100$). Isolines of $c_s/c_b^*$ values from the model and the pseudo-analytical solution are
presented on figure 9. To compute the pseudo-analytical solution, the reference level was chosen equal to $\delta z_b = 0.05H$ as in the work of Hjelmfelt and Lenau (1970) and the reference concentration is taken equal to $c_b^* = 0.025$ and applied as a boundary condition at the elevation $z = \delta z_b$.

Compared with the situation where the hypothesis on the flow are enforced (see figure 8), the model results do not match the pseudo analytical solution but the global behavior remains the same. Starting from no suspension, the suspended sediment
quantity gradually increases with the distance to the inlet until reaching an equilibrium situation where the settling and the turbulent diffusion cancel each other out. Figure 10 shows the vertical suspended sediment volume fraction $c_s$ profiles at different positions along the channel and shows the convergence toward an equilibrium solution close to a Rouse profile.

As stated previously the difference with the pseudo-analytical solution comes from the unrealistic hypothesis made to derive it. A better agreement could yet be found, for instance by playing with the boundary conditions for $\omega$ at the top and bottom
boundaries which would affect the shape of $\nu_t$ profile. Another adjustment parameter is the reference concentration at the reference level which is the bottom boundary condition of the pseudo analytical solution.

### 4.3 Idealized dune transport

As a first benchmark for the Exner equation (eq. 9), an idealized one dimensional dune transport model for which an analytical solution exists is presented. In this idealized case, a very simple flow is considered (see figure 11). The fluid is topped by a
rigid lid placed at an elevation $H$ from the bottom. The flow is considered vertically uniform with a constant discharge per unit width $Q$. The depth-averaged velocity is obtained by conservation of the mass, $U = Q/(H - z_b)$.

In this simplified case, only bedload transport is considered. To be able to derive an analytical solution of the Exner equation, the bedload $q_b$ must be expressed as a function of the bed elevation $z_b$. This is done by assuming that the bedload is a power



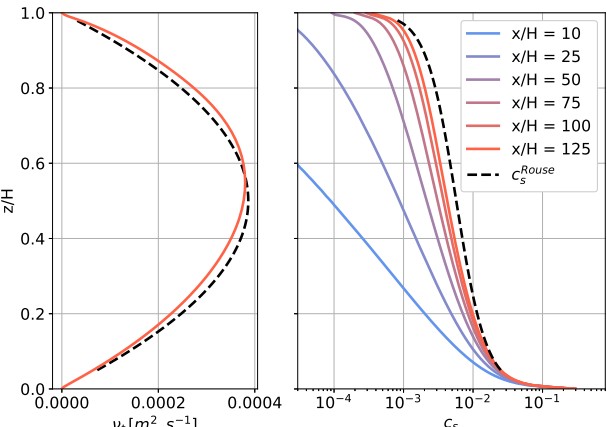

**Figure 10.** On the left hand side, the turbulent eddy viscosity obtained with the model (solid line) and the theoretical parabolic profile (black dashed line). On the right hand side, vertical profiles of $c_s$ at different x-positions in the channel. For comparison, the Rouse profile corresponding to the pseudo analytical solution on figure 9 is also plotted (black dashed line).

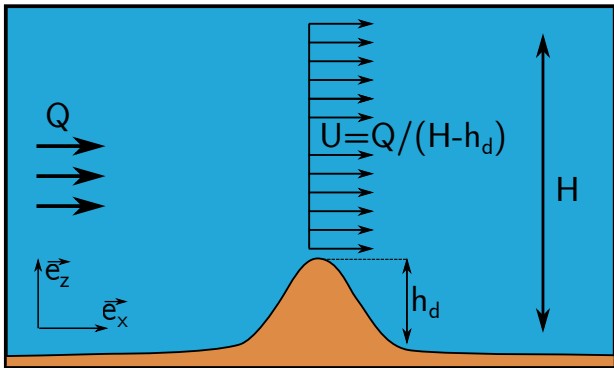

**Figure 11.** Schematic of the idealized dune transport case.

law of the depth-averaged velocity $U$, $q_b = \alpha_d U^{\beta_d}$, where $\alpha_d$ and $\beta_d$ are two positive constants. The Exner equation (eq. 9)
simplifies to:

$$\frac{\partial z_b}{\partial t} + c(z_b)\frac{\partial z_b}{\partial x} = 0 \, , \tag{27}$$

$$c(z_b) = \frac{\partial q_b}{\partial z_b} = \frac{\alpha_d \beta_d Q^{\beta_d}}{(H - z_b)^{\beta_d + 1}} \, , \tag{28}$$

where $c(z_b)$, is the celerity of the bed form. Starting with a given initial bedform $z_b(x, t=0) = F_0(x)$, the solution to equation
(27) is obtained with the method of characteristics leading to $z_b(x,t) = F_0(x - ct)$. Depending on $F_0$, shocks can occur as the
bedform migrates. A shock occurs if at least on some interval $\mathcal{I} \in \mathbb{R}$, the function $G : x \to c(F_0(x))$ is decreasing. The dune





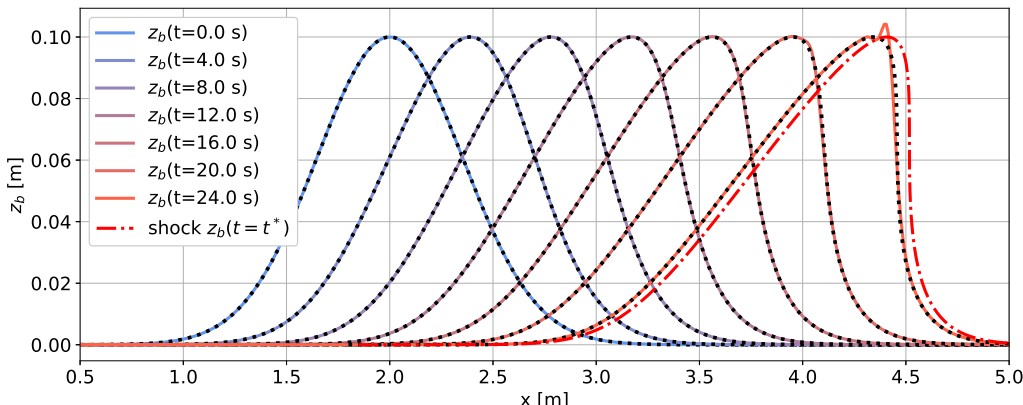

**Figure 12.** Dune transport problem, comparison between model results (solid lines) and analytical solution (black dotted lines) at different times.

celerity $c$ being an increasing function of $z_b$, shocks will occur if the initial bedform $F_0$ presents at least one negative slope. In this idealized dune transport case the initial dune profile is Gaussian:

$$F_0(x,t) = h_d\, e^{-\left(\dfrac{x-x_d^0}{\sigma_d}\right)^2}, \tag{29}$$

with $h_d$ the height of the dune, $x_d^0$ the initial position of the top of the dune and $\sigma_d$ a parameter linked to the dune width such that $F_0(x_d \pm \sqrt{\ln(2)}\sigma_d) = 0.5 h_d$.

With this initial dune profile, a shock will appear somewhere at the downstream side of the dune. In order to know the position and time of the shock, it is needed to find the position $x_0^*$ defined as follows, $G'(x_0^*) = min_{x \in \mathcal{R}}(G(x))$. It corresponds to the initial position of the point belonging to the caracteristic line on which the first shock occurs. The breaking time is then obtained as $t^* = -1/G'(x_0^*)$ and the shock position $x^*$ as well by advection of $x_0^*$ along its characteristic line, $x^* = x_0^* + G(x_0^*)t^*$.

The following parameters are chosen:

- flow properties, $H = 1\,m$ and $Q = 1\,m^2/s$

- bedload flux, $\alpha_d = 0.05$ and $\beta_d = 1.5$

- dune properties, $h_d = 10\,cm$, $\sigma_d = 0.6\,m$

For this configuration, the breaking time is $t^* = 24.67\,s$ and the shock position $x^* = 4.51\,m$. A solution to equation 27 is looked for between time $t = 0$ and the breaking time $t^*$. An Adams-Bashforth scheme of order 1 is used for time discretization and a linear upwind scheme for the advective term. A comparison between the model results and the analytical solution is presented in figure 12. Overall, the model fits well with the analytical solution except when the time get close to the breaking





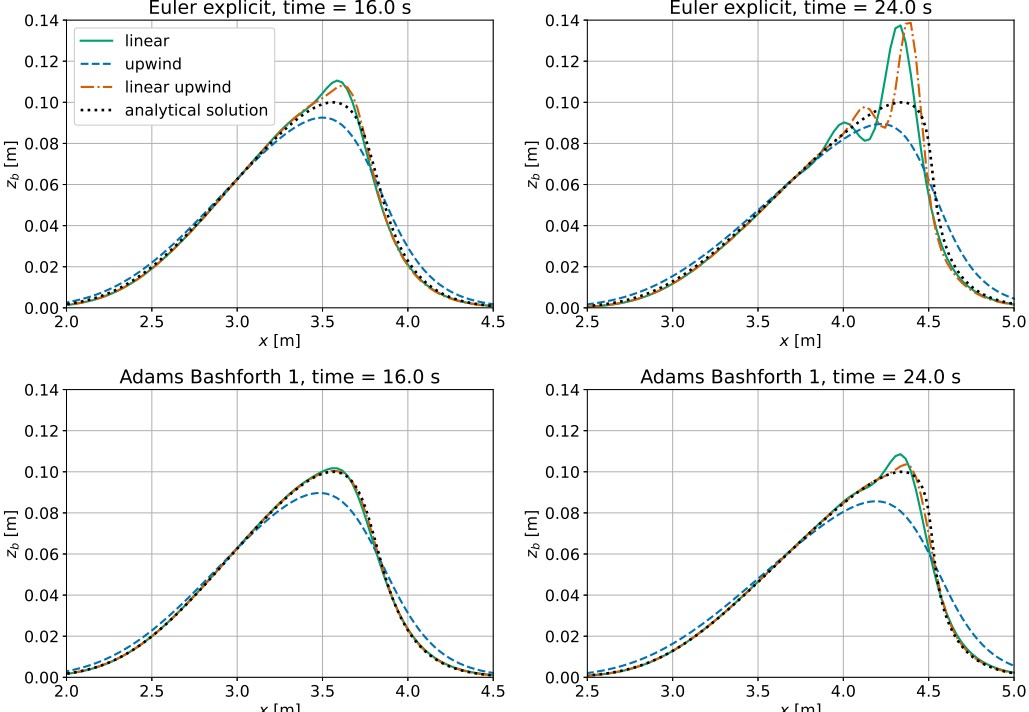

**Figure 13.** Comparison of results obtained with the three different numerical schemes for the advective term and the two schemes for temporal discretization (euler explicit on top plots and Adams-Bashforth 1 on bottom plots). The comparison is made at two different times, one intermediate time (left plots) and one time close to the breaking time (right plots).

time where a small instability is starting to develop. Figure 13 illustrates how the chosen numerical schemes affect the solution stability and precision.

At the dune front, the gradient of $z_b$ becomes important and consequently the gradient of $q_b$ as well. Depending on the numerical schemes used, this can trigger oscillations. Using the Euler explicit scheme for time discretization, the use of a second order scheme for advection leads to instabilities appearing on the crest of the dune. On the other hand, the low order upwind scheme brings up numerical diffusion and thus a poor prediction but ensures numerical stability. A better match between the numerical results and the analytical solution is achieved using a second order scheme for the temporal term

(Adams Bashforth 1) as the numerical solution no longer oscillates.

As stated in section 3.2, the interpolation from faces to vertices needed to enforce mesh motion acts as a filter, however, depending on the case, it may not be sufficient to suppress the appearance of numerical instabilities in particular in regions presenting steep slopes. The use of an avalanche model (eq. 13) brings more stability by limiting the maximum bed slopes. However it is not used in this example as no analytical solution can be derived for this problem if the avalanche mecanism is

taken into account.





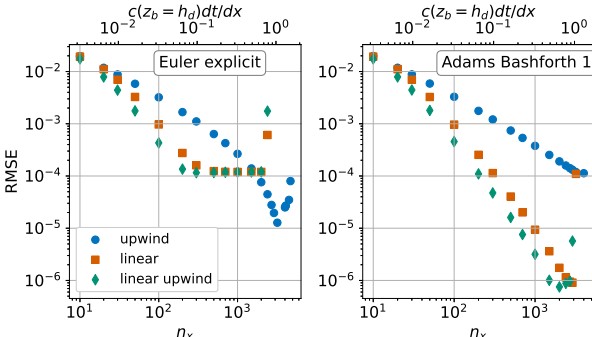

**Figure 14.** Root Mean Square Error for different combinations of schemes and mesh refinements. The time step is constant $\delta t = 0.05\,s$ for all simulations.

From the results presented in figure 13, the combination of a Adams-Bashforth 1 scheme for the time discretization and a linear or linear upwind scheme for the bedload flux, both being second order schemes, seems to offer the best compromise between stability and accuracy. Choosing an Euler explicit time scheme tends to trigger instabilities while the use of an order 1 upwind scheme leads to more stability at the cost of accuracy. To further illustrate the different behaviors of the possible scheme combinations and for different mesh refinements, multiple simulations are performed and the results are compared with the analytical solution.

The stability of the numerical solution is related to the mesh refinement through the maximum Courant number, whose evaluation is straightforward as the celerity of bedforms $c$ is known (eq. 27). The maximum Courant number is then $\max(C_o) = c(z_b = h_d)\delta t/\delta x$ where $\delta t$ is the time step value and $deltax$ the width of the mesh faces in the x-direction, the mesh being uniform. The accuracy of the numerical solution is evaluated using the Root Mean Square Error (RMSE):

$$\text{RMSE} = \sqrt{\frac{1}{N_F}\sum_f (z_b^n|_f - z_b^s|_f)^2}, \tag{30}$$

where the index $f$ stands for the finite area mesh faces, $N_F$ the number of faces of the mesh, $z_b^n$ is the elevation of face $f$ center (numerical solution) and $z_b^s|_f$ is the analytical bed elevation at face $f$ center. Each simulation is represented by a point in figure 14.

For low Courant number (poor mesh quality) all simulation are stables and present similar errors. As the mesh quality increases, the RMSE decreases but at a faster rate for second order schemes for bedload advection until the solution becomes unstable when the maximum Courant number gets close to 1. The sudden rise of RMSE values for high mesh resolution is the sign of those instabilities. The use of an upwind scheme allows to use an higher Courant number without the simulation failing. This is due to the numerical diffusion that this first order scheme involves. When using an Euler explicit time scheme along with one of the second order schemes for advection, it is observed that the RMSE does not depends anymore on the mesh resolution for values of maximum Courant number of 0.1 and higher until the appearance of instabilities. It shall be recalled here that a filtering process is applied on the numerical solution at each time step as the bed elevation increment is interpolated





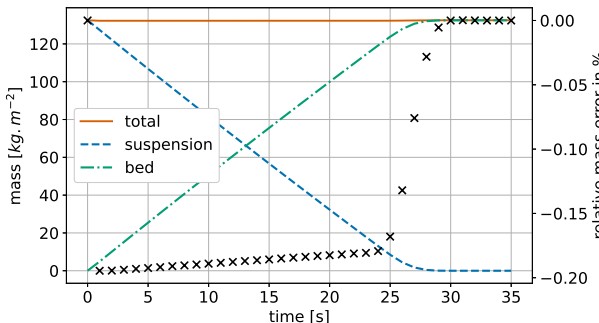

**Figure 15.** Variation over time of the sediment mass per unit area divided into suspended and deposited sediments. The relative error on the total mass in percentage is represented by black crosses.

from faces to vertices (discussed in section 3.2). The results presented support the use of a combination of an Adams-Bashforth 1 scheme along with a linear or linear upwind scheme to ensure both stability and accuracy.

### 4.4 Sediment settling

A still basin of depth $H = 1\,m$ is initially uniformly loaded with a volume fraction $c_s^0 = 0.05$ of suspended sediment corresponding to a mass concentration of $132\,kg.m^{-3}$. As the suspended sediment deposit, the bed level rises up and reaches a final elevation $z_{bed} = \frac{c_s^0}{1-\lambda_s}H$, where $\lambda_s$ is the porosity of the deposited granular material. The settling velocity being set to $w_s = 3.59\,cm.s^{-1}$, the time at which the last sediment deposit on the bed is $t = \frac{H-z_{bed}}{w_s}$. The variation over time of the sediment mass distribution between suspension and deposited sediments is represented on figure 15.

During a time iteration, the equation for the concentration of suspended sediments is solved and the erosion/deposition flux is computed and leads to a new elevation of the sediment bed boundary after resolution of the Exner equation (see figure 5). But the mesh motion being solved at the beginning of the time iteration, the bed level increment computed at a given time will only affect the mesh geometry at the next time step. As a result, there is a one-time step delay in the morphological response of the bed, which leads to an error in the total mass. However, this error returns to zero once all the suspended sediments have settled.

Another settling case is presented, this time a 2-dimensional case with non uniform settling. The computational domain is conic shaped, wider at the bottom (5cm) and narrower at the top (1cm).

Initially, only water is present in the domain and sediments are injected at the top boundary condition with a constant concentration. The sediments settle under the action of gravity and deposit on the bed. As the settling is non-uniform spatially, the sediment bed slopes get important on the extremity of the deposition mound and the avalanche mechanism takes over leading to the formation of a conic shape similar to the one observed in an hourglass. The repose angle is taken equal to $\beta_r = 32°$. The sediments diameter is $d = 0.29\,mm$ and their density $\rho_s = 2600\,kg.m^{-3}$. The sediment settling velocity $w_s^0 =$





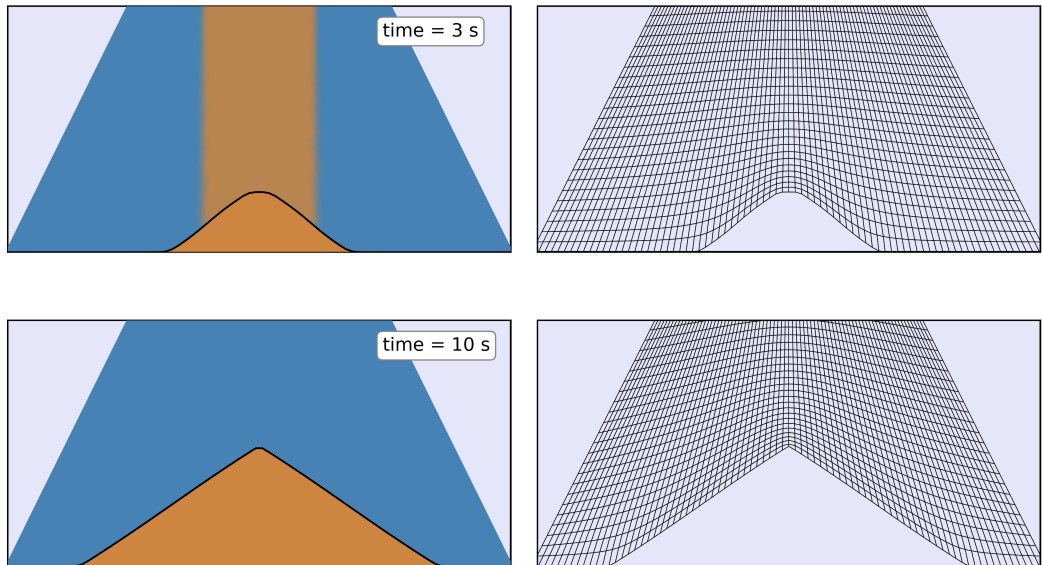

**Figure 16.** Representation of the sediment in the domain at two different times. At 3 seconds where a deposition mound is being formed as suspended sediment deposit and 10 seconds where all the sediments have settled. On the right part of the graph is shown the corresponding mesh to highlight the model mesh deformation capability.

$3.5\,cm.s^{-1}$ is computed using the formula from Fredsoe and Deigaard (1992) (see table 1) and is considered uniform as the hindrance effect is not taken into account. A constant flux of sediment is injected during 7 seconds by imposing a constant suspended sediment volume fraction at the top boundary $c_s = 0.05$. The simulation then run for 3 more seconds so that all sediments have settled by the end of the simulation (see figure 16).

The mass repartition of sediments between suspension and deposition is represented in figure 17. At the beginning of the
simulation, all the sediments are suspended and their quantity increases linearly over time until $t = 1.14s$ where the sediments start depositing on the bed. As the domain bottom boundary rises, the space occupied by suspended sediments shrinks leading to a diminution of the mass of suspended sediments. At 7 seconds, sediments stop being injected into the domain and approximately one second later, all the sediment have settled. Once again the mass error is evaluated by comparing the mass of sediment which has been injected into the domain to the sum of the suspended mass and the bed mass. Just as in the 1
dimensional case, the one time step delay in the bed morphology response induces an error on the total mass in the domain (see figure 17). This error then vanishes as the sediments settle so that the mass conservation is verified.





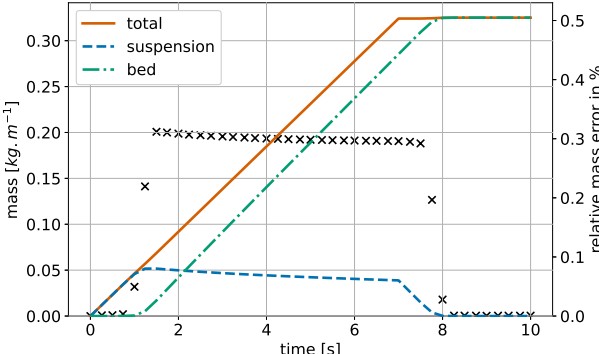

**Figure 17.** Variation over time of the sediment mass per unit area divided into suspended and deposited sediments. The relative error on the total mass in percentage is represented by black crosses. Suspended sediments is injected during the 7 first seconds and all deposit before the simulation end.

## 5 Application to dune transport

Numerous sediment transport problems involve bedforms of different sizes, from ripples to mega dunes, moving under the influence of the flow. As an example of application for *sedExnerFoam* the transport of a lone dune in a steady current is studied.

### 5.1 Configuration

The subject was studied experimentally by Kiki Sandoungout (2019) as he tried to identify different regimes of dune propagation and the dependence of those regimes to the flow conditions and to the dune mass. In this case study, the focus is made on one specific regime observed by Kiki Sandoungout and called the stationary regime. Two stages are observed, during the first one, the dune morphology rapidly changes from an initial conic shape obtained by deposition of sediments in still water. After some time, the dune reaches a stationary state and moves at a constant velocity in the flow direction.

The experimental facility consists of an hydraulic tunnel working in closed circuit with an experimental area made of a straight channel of length $L_c = 900\,mm$, of height $H_c = 90\,mm$ and of thickness $W_c = 6.03\,mm$. The flume is closed on the top by a rigid lid and is entirely filled with water. The granular material is made of glass beads of high sphericity. Their diameter is $d = 0.4\,mm$ and density $\rho_s = 2500\,kg.m^{-3}$. The particles terminal fall velocity in water is $w_s^0 = 7.67\,cm.s^{-1}$ which differs slightly from values obtained with the models presented in table 2 ($w_s^0 \approx 5\,cm.s^{-1}$). The friction velocity upstream of the dune is $u_* = 2.78\,cm.s^{-1}$ which corresponds to a Rouse number $R_o = 6.73$. This value indicates that bedload is the main transport mechanism for this problem. The critical Shields number ($\theta_c^0 = 0.079$) obtained experimentally is large compare to what is expected from the formulas in table 2. This could be due to the confinement of the particles in the flume, the ratio of the channel width to the particle diameter being only equal to 16.





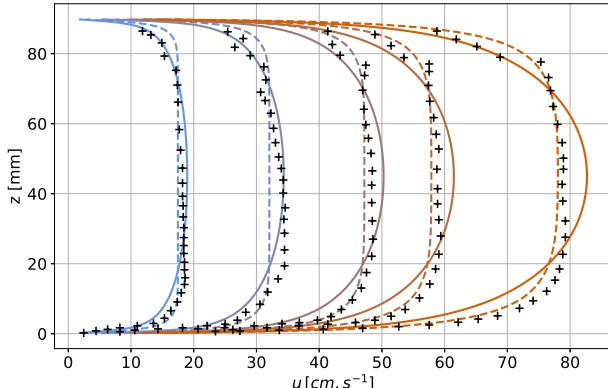

**Figure 18.** Velocity profiles obtained for different bulk velocities ($16.9\,cm.s^{-1}$, $30.9\,cm.s^{-1}$, $45.1\,cm.s^{-1}$, $54.9\,cm.s^{-1}$ and $73.2\,cm.s^{-1}$). The solid lines represent profiles obtained without friction on the lateral walls and the dashed line to the one obtained taking into account the lateral friction. The markers are experimental results from Kiki Sandoungout (2019).

An important aspect of this experiment is the thinness of the flume which makes the lateral wall friction not negligible. The lateral variation of the flow is neglected and the specific shear stress on the lateral walls $\boldsymbol{\tau_{wall}}$ is computed with the Darcy-Weissbach equation:

$$\boldsymbol{\tau_{wall}} = f\frac{|\boldsymbol{u}|\boldsymbol{u}}{8}, \tag{31}$$

where $f$ is the Darcy-Weissbach friction factor which can be computed explicitly with the equation from Swamee and Jain (1976).

$$f = \frac{0.25}{\left[\log_{10}\left(\frac{k_w/D_h}{3.7} - \frac{5.74}{R_{eW}^{0.9}}\right)\right]^2}, \tag{32}$$

where $D_h = 2H_cW_c/(H_c+W_c)$ is the hydraulic diameter, $k_w$ the roughness height corresponding to the roughness of the wall and $R_{eW}$ is the Reynolds number defined with the flume width. Integrating the momentum conservation equation (eq. 1) over

the flume width, a new source term $\boldsymbol{F_{walls}}$ corresponding to the effect of the lateral wall appears:

$$\boldsymbol{F_{walls}} = -f\frac{|\boldsymbol{u}|\boldsymbol{u}}{4W_c}. \tag{33}$$

The lateral walls effect on the flow is illustrated in figure 18, where vertical velocity profiles obtained with and without lateral friction are compared to experimental results for different bulk velocities. Lateral friction leads to a more uniform velocity field as the distance to the upper and lower walls increases, as well as to higher velocity gradients near the boundaries. The

agreement with experimental data is improved except at the top where the flow is disturbed by the presence of a screw hole,





which makes the data noisy and unreliable in this area. The main advantage of taking into account lateral friction as a source term is that it enables the use of a two-dimensional mesh, which significantly reduces the computational cost of the simulation.

A stationary regime configuration is reproduced numerically. A mass $m_0 = 10\,g$ of sediment is introduced through a hole drilled in the channel cover. It deposit under the influence of gravity and form a conic shaped mound with slope angles equal to the angle of repose of the granular material. Once the initial pile has formed, a motor is activated to create a left-to-right flow in the experimental zone with a bulk velocity $\overline{u} = 0.43\,m.s^{-1}$. Initially the sediments are loosely packed with a volume fraction of $c_s^{max} \approx 0.54$ in the bed. As the dune is transported by the flow the sediments get compacted and the sediment volume fraction in the bed increases resulting in the dune volume decreasing over time until the volume fraction reaches a constant value $c_s^{max} \approx 0.6$. This variation of the sediment volume fraction in the bed cannot be reproduced by the present model in which the bed porosity is considered constant over space and time. Therefore, it was chosen to initialize the dune with a volume corresponding to the one at the end of the experiment and not the initial one. As a result, the numerical dune is initially smaller than the experimental one but their volumes match after some time, once the granular material has compacted (see figure 20).

Regarding the boundary conditions, a uniform velocity is applied at the inlet $\overline{u} = 0.43\,m.s^{-1}$ and it was verified that the domain upstream of the dune was long enough for the flow to fully develop. Dirichlet conditions are also used at the inlet for the turbulent quantities, respectively $k = 0.001\,m^2.s^{-2}$ and $\omega = 15\,s^{-1}$ at the inlet boundary. It corresponds to a turbulent intensity $I_t = \sqrt{\frac{2}{3}k}/\overline{u} = 0.06$. Those values were chosen after simulating the flow in the flume without sediments and it was ensured that they did not impact the flow close to the dune. The top boundary is a rigid wall and a no slip boundary condition is thus applied on the velocity field. At the outlet, a zero gradient condition is applied to all field except the pressure for which a Dirichlet condition is used.

Experimentally, the fluid is initially still and the pump start to operate at $t = 0\,s$ accelerating the flow to the selected velocity setpoint. As the time it takes for the flow to accelerate and reach a mean velocity $\overline{u} = 0.43\,m.s^{-1}$ is unknown, it was chosen to initialize the problem differently. A first simulation of the hydrodynamics without morphological evolution runs for 10 seconds until the flow over the dune reaches a steady state. The morphological evolution is then activated and the dune begins to move under the influence of an already fully developed flow. The results are illustrated in figure 19 which represent the dune and the flow of water at three different times.

This inconsistency in the initial condition leads to a different morphological response in the first few seconds of the simulation. The different adjustment parameters of the model were thus tuned to match the experimental results beyond the first 5 seconds of simulation. As seen on figure 20, the numerical and experimental dunes are not matching the experiment during the first few seconds but as they approach a stationary regime their shape and velocity start to align satisfactorily.

To obtain the results presented in figures 20 and 21 multiple attempts were made and a sensitivity analysis to the different model parameters has been conducted. A first element that significantly affects the results is the resolution of the mesh and in particular the near bed resolution in the areas where the flow is highly non uniform. In this case study, it corresponds to the upstream slope of the dune where the flow is contracted and accelerated until the top of the dune where the flow detaches





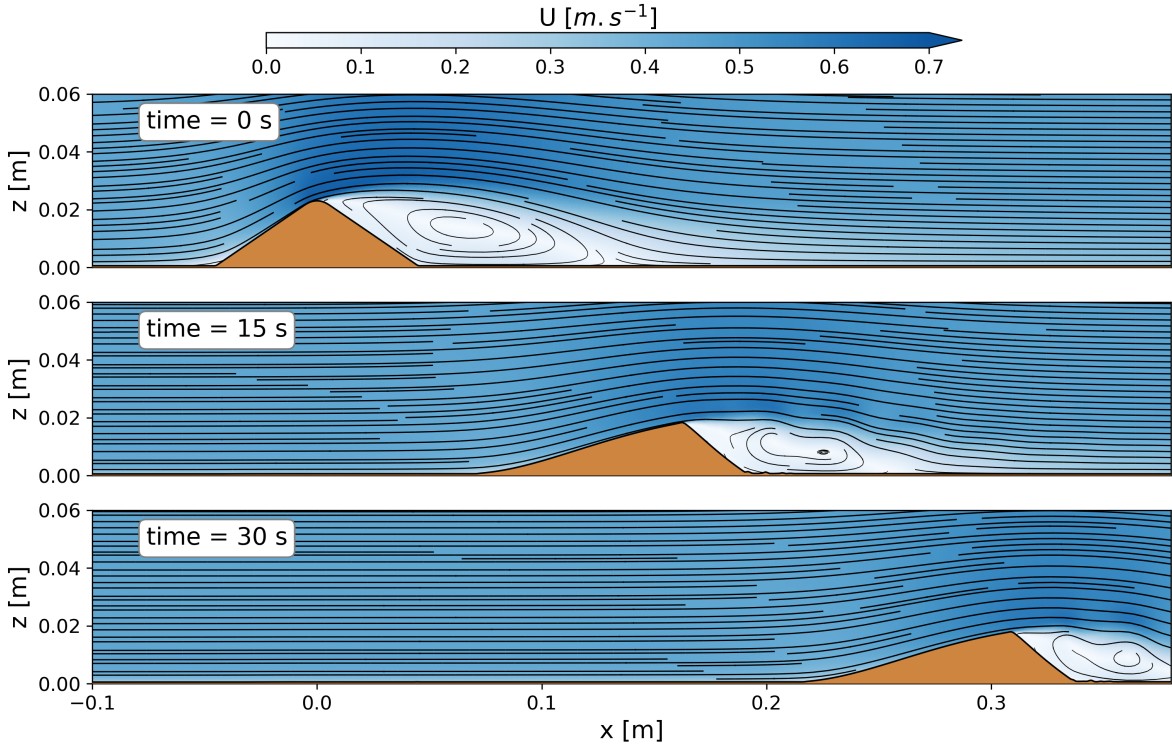

**Figure 19.** Representation of the dune and the flow streamlines at different times during the migration process.

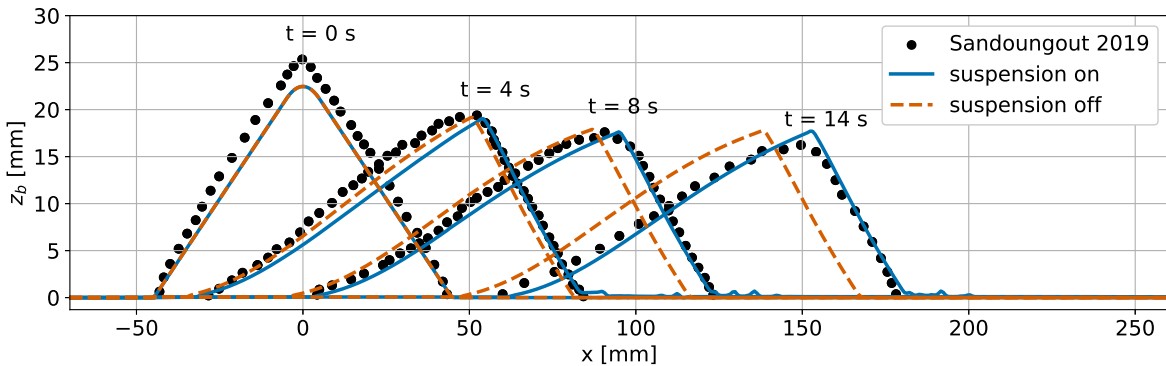

**Figure 20.** Sediment bed elevation at different times, 0, 4, 8 and 14 seconds. Comparison between results from two simulations and experimental results. The simulations use the same mesh, same parameters but for one, the suspended load is not taken into account.

and generates a recirculation cell illustrated in figure 19. A poor near bed mesh quality leads to an under estimate of the bed shear stress and as a consequence to a slower dune migration. The distance in wall unit between the cell centers making up the first layer of cells above the bed boundary and the bed boundary is kept between $z^+ = 1$ and $z^+ = 5$. Not surprinsigly the





numerical scheme used to discretize the advective term in the momentum equation (eq 1) was also found to affect the dune shape. This is mainly because of its effect on the recirculation cell caracteristics and the position of the detachment region which is located upstream of the dune crest with high order schemes but downstream with a low order *upwind* scheme. When the flow detachment appears downstream of the dune crest, the dune shape was found to take a rounder shape not matching the experiment. A second order *linear upwind* scheme was used to produce the results presented.

Another parameter of importance in this case is the critical Shields number $\theta_c^0$ which is evaluated in the range $0.033-0.046$ by the different models presented in table 2 but was experimentally estimated at a higher value of $0.079$. Increasing the value of $\theta_c^0$ not only slows down the dune but also reduces its height and length in the stationary regime. An intermediate value of $\theta_c^0 = 0.05$ was found to yield good results. In addition the critical Shields number was corrected with the local slope according to equation 11. A last key parameter is the formula chosen to calculate the bedload transport. It was found that the formulas presented in table 2 were all predicting dune velocity at least two times slower than the one observed experimentally. Therefore, a custom bedload formula $\phi_b = 32\theta^{1/2}\varpi(\theta-\theta_c)$ was used. It corresponds to an intensified version of the formula from Nielsen (1992) and can be considered reasonable in view of the significant scatter associated to bedload measurements (Recking, 2010). Indeed, even if these formulations are commonly used to model sediment transport in a variety of flow conditions, they are empirical relations derived from data of uniform flows in a straight channel. Therefore, they may not precisely describe sediment transport in accelerated flow regions, recirculation cells and other non uniform flows features.

A last critical point is the inertia of the sediments. On the upstream slope of the dune, the flow is accelerated and the bed shear stress increases. At the position where the flow detaches, the shear stress value suddenly drops. If the inertia of the bedload is not taken into account, then the sediments accumulate at the crest and the dune height increases. At some points the upstream slope becomes steeper than the angle of repose and the avalanche bedload compensates the shear induced bedload. The dune then stops moving and stays stuck in a non physical state. In reality, the sediments arrive at the crest with a certain velocity and some distance is needed for them to react to the sudden change of the bed shear stress. They could even be launched into suspension due to the abrupt change of slope at the crest of the dune. To retrieve a behavior of the dune migration similar to the experiment, it was found necessary to consider the sediments inertia which is done at first order by using the saturation of the bedload transport (see eq. 14).

The dune morphological parameters over time which are the dune position, its height and length are represented in figure 21. The dune position is represented by the coordinate $x_h$ located at midheight on the downstream slope of the dune. The dune height and length are estimated from the base of the triangle formed by two straight lines fitted on the dune upstream and downstream slopes. Two simulations with and without considering the suspended load transport are represented. As expected because of the high value of the Rouse number ($R_0 > 6$), the suspension is having little effect on the dune evolution. Its height and length in the stationary regime remain unchanged and regarding the migration velocity only a small difference is observed, $9.88\,mm.s^{-1}$ and $8.75\,mm.s^{-1}$ for simulations with and without considering the suspended load respectively. This velocity difference is better observed in figure 19 showing the bed elevation profiles at different times. For the case with suspension, a part of the suspended sediments passes over the dune and settles in the recirculation cell forming the small piles observed. They are then taken up by the dune as it migrates.





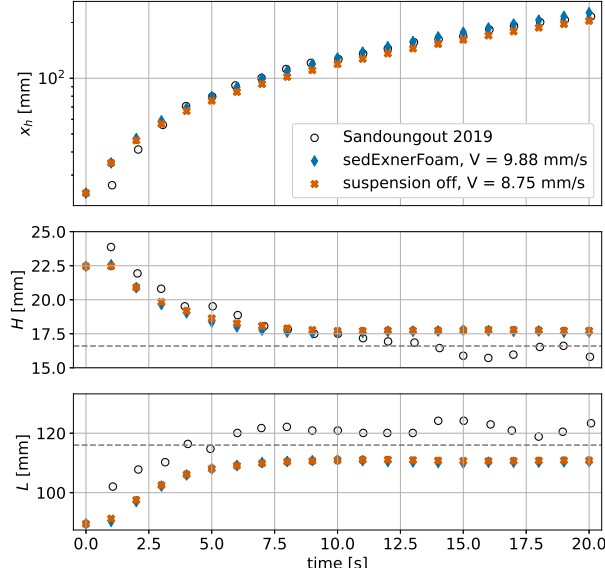

**Figure 21.** Dune morphological parameters evolution in time. From top to bottom are plotted the dune position represented by the coordinates $x_h$, which is located halfway up the downstream face of the dune, as well as the dune height and the dune length.

Overall the numerical dune is reproducing the experimental one well. But some discrepancies are still observed. The crest
of the dune is sharp in both numerical simulations compared to the experience and as a result, the height of the dune is slightly overestimated (see figure 21). At the same time the length of the dune seems to be underestimated but it could also be a consequence of the method used to estimate the dune length. The two stages of the dune migration are clearly observed. During the first 10 seconds, the dune shape changes rapidly, its height decreases and its length increases. At 10 seconds, the dune has reached a stationary stage and its shape remains unchanged as it migrates at a constant velocity.

**6  Conclusions**

A new numerical code, *sedExnerFoam*, aimed at studying sediment transport and the evolution of morphology, is proposed. Developed within *OpenFOAM®* (v2412), it is based on the *pimpleFoam* solver. Numerous closures for the settling velocity of particles, the bedload flux and the critical Shields number are implemented and can be modified by the user, thanks to the object-oriented environment offered by *OpenFOAM*.
The model has been extensively validated using multiple tests against analytical solutions or experimental data, covering everything from channel suspension to idealised dune transport, sand deposition, and mass conservation in an hourglass. These benchmarks were selected to isolate and test each component of the model individually. Lastly, applying the model to the migration of a lone dune under the influence of a steady flow illustrates its capability to handle complex problems. This process involves flow detachment, avalanching and bedload flux saturation, and is associated with significant mesh deformation. As



the position of the flow detachment point is particularly important, the turbulence model and the numerical scheme used to discretize the advection term in the fluid momentum equation must be chosen carefully. These choices can significantly affect the flow separation and the underlying morphodynamics. It was also found that the inertia of the sediments transported as bedload is essential for describing the sediment fluxes at the crest of the dune in order to match the morphological evolution observed in experiments.

Various future developments are being considered to overcome the model's current limitations. One such development is the implementation of a free surface, which is not currently taken into account. Another important limitation is the level of mesh resolution required for accurate estimation of bed shear stress in regions of non-uniform flow. This has been found to be systematically underestimated in RAS simulations without a very fine grid. To apply this model to problems on the scale of actual hydraulic structures, such as bridge piles, weirs or dams, new closures that are less sensitive to grid resolution

must be developed and implemented. Currently, the model is also limited to non-cohesive, monodisperse sediments and flows with relatively low suspended concentrations, as it does not take into account the feedback of the suspended load on the hydrodynamics. Future developments to the model could include implementing models for the transport of cohesive sediments, such as mixtures of silt and clay that are commonly found in estuaries. Alternatively, a framework could be developed to handle multiple classes of sediments of different sizes and densities. Additionally, *sedExnerFoam* could be employed alongside two-

phase flow models, such as *sedFoam* (Chauchat et al., 2017), to derive more accurate and robust closures for sediment transport fluxes through an upscaling process.

*Code and data availability.* *sedExnerFoam* (Renaud et al., 2025) model code, associated librairies, tests and tutorials are all available via zenodo at https://doi.org/10.5281/zenodo.15535485 or directly via github at https://github.com/SedFoam/sedExnerFoam. Instructions for installation and explanations on the repository organization are provided in a README file.

*Author contributions.* Author contributions. JC, CB, and OB designed the project. MR developed the source code, ran simulations, and wrote the paper. JC edited the manuscript. Supervision: CB, OB, JC. All authors discussed the results and contributed to the final paper.

*Competing interests.* The contact author has declared that none of the authors has any competing interests.

*Acknowledgements.* This work was carried out within the framework of the Oxalia Chair supported by the Grenoble INP Foundation. Various graphics presented in this work were produced using the Python package *fluidfoam* (Bonamy et al., 2025).



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
