# Peer review of "sedExnerFoam 2412: A 3D Exner-based sediment transport and morphodynamics model"

_EGUsphere, 2025_

## Referee Comment (RC3)

Overall, it is a promising paper that is worth publishing as it can be a useful tool for many civil engineers. The manuscript does well in explaining some chosen test cases to explain the model performance. However, I think there are some points worth of attention and major revisions are necessary before final publication.

**General comments**

G1. The model shows no three-dimensional results and performance while this is computationally different. Especially the introduction talks about processes such as scour (lines 41 – 49 and figure 1), which is then not covered as a complete case for the remainder of the manuscript. I think these three-dimensional cases are worth showing to truly trust the model.

G2. The style of writing of the paper concerns me. To me, it reads like a textbook for students or like an informal review paper describing the history of development of morphodynamic modelling. Overall, the style should be changed and the quality of English should be improved (examples of which are provided in the Technical comments below).

Specifically, I recommend to either (1) add the historical framing to the aim of the overall paper, or (2) remove the historical context as it does not contribute scientifically. Some examples:

   a. Lines 76-83 mention all kinds of different turbulence closures, but in the end only one is implemented.

   b. Line 122: "...an Austrian meteorologist and geophysicist." Not sure why this is relevant.

   c. Line 135: "In the 1930's, Albert Frank Shields made measurements of the motion threshold already highlighted by Du Boys in 1879." Vague and misses citations

   d. Lines 161-165. Unclear why alternatives to modelling avalanches are mentioned when a different implementation is used.

   e. Section 2.4.3 provides a lot of context and alternatives, while it seems that only one implementation is done in the end (even mentioning "In *sedExnerFoam* it was chosen to avoid the use of two different meshes" in line 203). Focus on this formulation and avoid confusion.

G3. Model performance, convergence and computational costs are missing to the manuscript and how this compares to other morphodynamic models. Where should this newly developed model be framed, given the morphodynamic models discussed in the introduction?

G4. To me, a separate discussion of the weak points of the developed model is missing. I do notice these kinds of points returning in the main text and the conclusion, and I understand that the migrating-dune case is considered the benchmark, but a separate discussion would be appropriate, such that the conclusion can focus on a fundamental summary.

**Specific comments**

S1. The abstract should be rephrased, it currently has no conclusion.

S2. Line 41: Scour is explicitly covered in the introduction, but later, no reference is made to it, and it's not a use case. Furthermore, it's a 3D process, and it is unclear whether the newly developed model is able to achieve this.

S3. The introduction misses a clear aim.

S4. The end of the introduction should clearly state the structure of the paper with corresponding sections.

S5. Lines 109 – 114. Not sure why all of this is mentioned while a constant value is assumed in the end.

S6. Line 130: D and E are not fluxes, but a gradient of fluxes (just like q_b is a flux).

S7. Line 141: Equation 10 has no reference to it and no context or explanation.

S8. Line 156: Equation 12, and in general, unclear what omega is.

S9. Line 156: Equation 12 could be removed, as it is just one of the many sediment transport equations available to the user, listed in Table 2.

S10. Line 167, Figure 4: Not sure whether it is worth it to show this figure, as they are not your results.

S11. Line 175: Equation 14 is one-dimensional, while bedload q_b is in the text considered a 2D flux.

S12. Line 354 & 399: unclear what $z^+$ is.

S13. Line 365: "For all 4 simulations, the turbulent Schmidt number was set to values slightly above 1, $\sigma^c \in [1.1, 1.2]$" Why is this done? Is this part of a calibration step that is not described?

S14. Line 369 talks about "adjusting"; is calibration meant by this?

S15. Lines 384-385: "Rouse number is equal to 0.5 which corresponds to a highly suspended regime." Is this still valid concerning the assumption that suspended load does not influence hydrodynamics, as mentioned in line 64?

S16. Lines 456-461: I don't understand which numerical schemes are implemented here. I understand the Euler explicit time scheme, but I'm only familiar to the notion of an upwind scheme when dealing with spatial discretisation of advective terms. In conclusion, I do not understand how a temporal scheme and a spatial scheme are compared here. Could you explain further?

S17. In addition to the point made above; the evaluation of the different numerical schemes is not appropriate within Section 4: Validation.

S18. Line 482: I am not an expert on suspended sediments, but 132 kg/m3 seems like a very high concentration and seems to violate the assumption of dilute sediments only (line 64). Could you comment on this?

S19. Section 4.4: The presented cases are all purely deposition cases. The time-step delay in the morphological response (lines 489-491) indicates that mass is not conserved. This is not a problem in the example given, but could you comment on how this influences more realistic cases when hydrodynamic conditions are more complex?

S20. Section 5.1: A lot of calibration seems to be necessary to get representative results. Especially, the critical shields parameter and the bedload transport formula need to be greatly adapted to get proper model validation. How do you reflect on the validity of the results? Are the results shown essentially a "model overfit" for the specific experiment

examined, or can the same parameters also be used to model different experiments in the same flume from Sandoungout (2019)?

S21. Section 5.1: The results would become a lot stronger if a validation of the hydrodynamics over the dune is done as well.

**Technical comments**

T1. General technical 1: please use more comments to improve readability. Examples:
   a. 408: As stated previously the difference with the pseudo-analytical ...  -> As stated previously, the difference with the pseudo-analytical ...
   b. 442: Overall, the model fits well with the analytical solution except when the ...-> Overall, the model fits well with the analytical solution, except when the
   c. 592: In addition the critical Shields number ...-> In addition, the critical Shields number

T2. General technical 2: The citation format used (especially the use of first names) is inconsistent and not in line with the journal's expectations.

T3. Line 32: ALE is mentioned but not what it means

T4. Line 89: Equation 5 has weird formatting

T5. Line 109: "... hot topic today ...", informal -> Rephrase

T6. Line 131 – 131:  "2 dimensional" -> "two-dimensional"

T7. Line 144: "Accurately measuring the threshold of motion is a difficult task mainly because of the absence of a universal definition of the motion threshold." Vague and unclear -> rephrase

T8. Line 167: adimensioned -> dimensionless

T9. Line 172: "eolian", unclear what this is.

T10. Line 181: "hard point", informal and vague -> rephrase

T11. Line 190: "for a lot of situation", vague and informal -> rephrase

T12. Line 192: "to handle out of", unclear -> rephrase

T13. Line 238: "split" -> "discretised"

T14. Line 265-267: "The users have the choice to use either an explicit Euler scheme or a first order Adams-Bashforth scheme, which is a second order time scheme, for temporal discretization." Confusing -> rephrase

T15. Line 273-275: "Jacobsen (2015) made a detailed review of the different possible methods to solve the Exner equation and analysed their benefits and shortcomings. He proposed a mass conservative interpolation scheme which is the one implemented in the current model." Informal, rephrase

T16. Lines 314-325: Rephrase, informal use of English and many redundant words.

T17. Line 375 & 381: "hypothesis" -> hypotheses

T18. Line 398: vary -> varies

T19. Line 440: "looked for", rephrase

T20. Line 442: get -> gets

T21. Line 464: deltax -> \delta x

T22. Line 470: "all simulation are stables" -> "all simulations are stable"

T23. Line 485: " the sediment bed slopes get important on the extremity of the deposition mound" -> rephrase